# Influence of Emotions on Elementary-School Children’s Implicit Learning during a Serial Reaction Time Task

**DOI:** 10.3390/children10030533

**Published:** 2023-03-09

**Authors:** Mélanie Mazars, Aurélie Simoës-Perlant, Pierre-Vincent Paubel, Céline Lemercier

**Affiliations:** Laboratoire CLLE CNRS UMR5263, Université de Toulouse 2 Jean Jaurès, 31058 Toulouse, France

**Keywords:** implicit learning, children, SRT task, emotions

## Abstract

As pedagogues and childhood professionals, teachers must be aware of both implicit and explicit learning processes. They must also bear in mind that learners’ performances may be influenced by the many emotions triggered by different situations at school (e.g., fear of failing, happiness upon succeeding, anger at leaving work unfinished). The objective of the present study was thus to analyze the efficiency of implicit learning among 8- to 11-year-old children and the impact of emotions on this type of learning. In order to analyze implicit learning, 65 elementary-school children performed a serial reaction time task in a laboratory context. Emotions were induced by asking the children to read six short emotional sentences and listen to classical music. Results showed a significant impact of the task condition (semi-random or fixed sequence) on reaction times. Moreover, the induction of happiness resulted in slightly longer reaction times compared with neutral induction. These results need to be deepened to better understand the interactions between emotions and implicit learning in children.

## 1. Introduction

Teachers help children go through all their developmental stages and build their academic and social skills. They must therefore understand both explicit and implicit learning processes to enhance pupils’ learning abilities. Implicit learning is present every day at school. In reading activities, the implicit identification of words [1] helps pupils read faster and focus on understanding text. Children use implicit orthographic rules [2] during spelling tests or writing activities. In sport, implicit motor actions permit children to focus on intentional strategies to succeed [3]. However, school situations can trigger a variety of emotions in children (e.g., fear of failing, happiness upon succeeding, anger at leaving work unfinished). These emotions may impact implicit learning and pupils’ abilities to succeed at school. Teachers must therefore have a thorough knowledge of learning processes and how emotions may influence them. The purpose of this research is to take a position on the existing debate regarding the impact of emotion on implicit learning in children.

### 1.1. Implicit and Explicit Learning: Definitions

Two types of learning processes can be identified: implicit learning and explicit learning. Explicit learning is intentional learning. Students can plan, identify, and verbalize it. For Rebuschat and Williams [4], explicit learning involves “learning scenarios during which people actively observe patterns; in other words, they learn intentionally. This process leads to conscious knowledge.” (p. 829). Seger [5] defined implicit learning as “a non-episodic and an unintentional learning of complex information” (p.163). Thus, implicit learning is an unplanned and unintentional process that requires attention [6]. Its skills are indirectly identified. In this type of learning, individual behaviors gradually adapt to the situation. Nevertheless, in reality, these two ways of learning are not so easily identifiable. In daily living (e.g., reading, doing sport, listening to music, language learning), implicit and explicit learning are used in a complementary fashion to develop expertise [1,3,7].

### 1.2. Reber’s Influence

Two main methods have been used by researchers to investigate implicit learning. One of the first paradigms to emerge in the literature, which has been the origin of a great deal of research, is the artificial grammar paradigm [8]. In the princeps paradigm, Reber created a two-phase procedure: an artificial grammar learning phase and a test phase. During the grammar learning phase, participants have to memorize consonant sequences. These sequences follow artificial grammatical rules that are unknown to the participants. During the test phase, participants have to memorize other sequences of letters, half of which follow the artificial grammatical rules. Results show that participants are better at memorizing sequences of letters following the artificial grammatical rules. Moreover, they are able to say whether or not these sequences of letters correspond to the artificial grammatical rules without explicitly knowing them. Reber concluded that people can implicitly learn new grammatical structures and then identify them in other letter sequences. The work of Reber served as a theoretical foundation in the field of implicit learning, and he defended the idea of the primacy of unconscious and implicit functions over explicit conscious functions and proposed a series of hypotheses, according to which implicit learning would be (1) independent of the age of the subjects and of the level of development; (2) not very variable between individuals; (3) independent of the intellectual level; and (4) robust in the face of pathology. 

The second main method involves Nissen and Bullemer’s serial reaction time (SRT) task [9]. In this task, a visual target (e.g., word, image, symbol) moves between four rectangles on a computer screen. Each rectangle is associated with a different key on the computer keyboard. Whenever the target appears in one of the rectangles, the participant must press the corresponding key as quickly as possible. The movements of this visual target follow fixed or semirandom sequences. These sequences are therefore determined by the examiner, but the participants are not aware of them. Implicit learning is attested if the individual shows quicker reaction times and lower error rates for repeated sequences than for semirandom ones. These paradigms are similar on some points (e.g., subjects are confronted with an environment structured by a set of more or less complex rules); but they differ on others (e.g., what is measured may or may not refer to what is explicitly learned by the subjects; the degree of abstraction is not identical from one task to another). The consequence of these differences is an imprecision in the very notion of implicit learning.

### 1.3. Criticisms of Implicit Learning Research

Since the 1960s, there has been a steady rise in the number of implicit learning studies based on these methods (from 13 articles between 1960 and 1970 to 1907 articles between 2000 and 2010; to 3450 articles from 2011 to 2020, MEDLINE). One criticism of these studies is that most of them were conducted with (1) adults and (2) in a laboratory context [10]. In a more ecological context, there have been only a few implicit learning studies with children (for reading activities [1,2,6], for grammatical spelling [11], for the acquisition of liaison in oral language). They all reveal the presence of implicit learning, particularly for children in both oral and written language. For example, Pacton and Perruchet [12] describe the links between implicit learning and the school context. For them, implicit learning sometimes influences children’s answers (e.g., plural marks), even if these answers do not correspond to the grammatical rules learned at school. Moreover, Pacton and Perruchet suppose that implicit learning begins early, even before school starts. Recently, some studies have indeed focused on early implicit learning skills. McLeod and MacDade, and then Dickinson et al., show that preschool children can learn vocabulary implicitly through storytelling [13,14]. In mathematics, Moore et al. also show that kindergarteners’ implicit understanding of arithmetic helps them process symbolic numerical magnitude [15]. These studies corroborate the early aspect of this type of learning, well before the child enters school [16]. To summarize, implicit learning has been revealed in a laboratory context in adults as well as in children, but it has also been demonstrated in more ecological studies conducted in a school context. Although implicit learning is considered a robust phenomenon, it has been linked to the emotions felt by the subjects.

### 1.4. Emotions and Learnings

While the effects of emotions on explicit learning are well established (for a review, see [17]), the effects on implicit learning are less documented. A few studies have addressed this phenomenon through the use of the SRT paradigm. Two of these studies investigated implicit sequence learning with depressed patients [18,19]. The results of these studies showed significantly poorer performance than control subjects. The study by Shang, Fu, Dienes, Shao, and Fu corroborates these early findings by implementing an emotional induction protocol using music followed by a sequential reaction time task in which adult participants had to track a target of different shapes (square, triangle, circle, and heart) and colors (red, green, yellow, and blue) as quickly as possible [20]. The shape/color association was governed by certain statistical rules, and it is on the basis of this implicit construction that the reaction times of the subjects had to decrease significantly compared with those in the randomized condition. Several methodological elements should be considered in this study, notably the emotional induction tool chosen (i.e., music) and the fact that color can also induce emotions. Among the tools of emotional induction, music is particularly interesting in the sense that we are able, even without being musicians, to distinguish sad music from happy music in half a second of listening [21]. Similarly, this ability to distinguish between sad and happy tones is thought to be operational in children as early as 3 years of age [22] and is thought to improve with development. The method of inducing emotions through a musical medium did not have a standardized method until the work of Vieillard, Peretz, Gosselin, Khalfa, Gagnon, and Bouchard, who developed a set of 56 musical sequences to induce positive and negative emotions [23]. Shang et al. chose this means of induction but did not, in their first study at least, evaluate the effects of a “neutral” emotion [20]. Yet, based on the attentional resource allocation and cognitive interference model [24], emotion, whether positive or negative, would have a deleterious impact on cognitive performance. Thus, to overcome this difficulty in their second study, the authors added a control group (under neutral emotion) induced by reading a text. Here again, the neutrality of the material is to be considered experimental since the induction material is different between groups (music vs. text). Furthermore, the results of Shang et al. highlight weaker performance under sad emotion than under happy emotion in the implicit sequence learning task when the statistical regularities are on the shapes [20]. On the other hand, when the statistical regularities concerned colors, the authors were unable to identify any difference in performance. One avenue of analysis, which has not been mentioned, would be to consider color as an element conveying emotions [25]. 

The results of the three studies using the SRT task suggest that a negative emotional state may have a deleterious impact on implicit sequence learning abilities in adults. A study conducted by Kwok seems to qualify this type of conclusion [26]. Indeed, after inducing a happy or sad emotional state in a group of typical adults, the authors proposed a sequential encoding task during which the subjects, when confronted with two superimposed images (one with negative emotional content and one with neutral content) for a very short period of time, had to inhibit the negative image and verbalize the neutral image. In a second phase, a set of negative images was presented to them, and they had to say whether or not this image had been presented during the encoding phase. Kwok’s hypothesis is based on theories relating to emotional congruence, according to which negatively induced subjects should perform better on the task of recognizing negative images (i.e., whose valence is therefore congruent with their emotional state) than positively induced subjects [27,28,29,30]. However, the results do not seem to support his hypothesis since he found undifferentiated performance regardless of the group of subjects. Similar results were obtained by Bertels, Demoulin, Franco, and Destrebecqz with the use of a statistical visual shape learning task [31]. Negatively induced participants did not differ on the basis of their performance from neutrally induced participants. In line with these results, the study conducted by Sentman does not show any significant difference in performance on the sequential reaction time task according to the emotional state of the subject [32]. On the other hand, in a more interesting way, the author shows differences linked to emotion in the artificial grammar task, with implicit learning performances that seem to be better under negative emotion. This result is consistent with the findings of Cabral, Wilson, and Miller’s [33] meta-analysis that motor performance (in the sports domain) in an implicitly learned task is better when participants feel high pressure. The rationale for these results is that implicitly learned skills can be performed with greater automaticity (i.e., fewer attentional resources), thus allowing performance to be maintained when the felt emotion leads to a misallocation of attentional resources to external stimuli.

Despite the limitations mentioned above, these studies shed new light on the question of the specificity of implicit learning. While performance on many explicit tasks can be influenced by the emotional state of the subject, the impact of emotion on implicit learning is more nuanced. As the implicit system is less dependent on attentional resources, it should also be less affected by the emotions felt by the subject during the task. Thus, differences in performance according to the emotion induced should be highlighted not only in the comparison of explicit vs. implicit knowledge tasks but also in the comparison of implicit learning tasks of different natures.

The few studies that have attempted to evaluate the impact of emotional state on implicit learning abilities have yielded contradictory results. Here again, the analysis of methodologies allows us to assume that this type of learning could be dependent on the characteristics of the task and the subjects. Sentman’s study compares the performance of adult subjects induced positively, negatively, or neutrally on two implicit learning paradigms, which, as mentioned earlier, do not mobilize the same skills [32]. The artificial grammar task requires the extraction of probabilistic information, whereas the SRT task relies on learning specific sequences. More specifically, assessing implicit learning of sequences under different modalities, conveying or not conveying an emotion (i.e., using for example tracking of colored targets or passive listening to music), would allow us to apprehend the cognitive cost of this type of skill during development. To our knowledge, no such study has been conducted on children.

### 1.5. Objectives and Hypothesis

The purpose of this research is to take a position on the existing controversy about the impact of emotion on implicit learning. To date, contradictory results emanate from the literature, and to our knowledge, this effect has not been evaluated in children. Thus, we subjected children aged 8 to 11 years to an SRT task after having induced them emotionally via two inducers. Based on the work of Ellis and Moore [24], we assume that emotion will have a deleterious effect on implicit sequence learning.

## 2. Materials and Methods

### 2.1. Population

We recruited 69 fourth and fifth graders in several public elementary schools (Occitanie region, south of France). The data of four participants were excluded from the analysis because they declined to complete the SRT task. The 65 remaining participants (41 girls and 24 boys), aged 8–11 years (mean age = 9.69 years, SD = 0.7), were divided into two subgroups: induction of a neutral emotion (n = 27; 15 girls and 12 boys, mean age = 9.88 years old) and induction of happiness (n = 38; 26 girls and 12 boys, mean age = 9.55 years old). 

### 2.2. Material

#### 2.2.1. Serial Reaction Time (SRT) Task

The SRT task used in the present study was designed by the CLLE laboratory (UMR 5263). It had previously been tested with about twenty children. This computer-based activity consists of following a target (star) as it moves from one white rectangle to another. There are 4 rectangles (A, B, C, D) on the screen. The task is composed of five blocks of 54 trials each. These 54 trials alternate between 30 displacements for fixed sequences and 24 semi-random displacements for semi-aleatory sequences (for more details, see Appendix A). A different sticker is placed on each of the four keys on the keyboard to make it easier for participants to identify them. The software automatically registers reaction times (i.e., time between target appearance and keyboard pressure) for each trial in milliseconds. It also measures whether the participant presses the right key (1) or the wrong key (0). The software does not wait for a correct response before switching to the next trial. This kind of SRT task is often used to highlight implicit sequential learning [9]. 

#### 2.2.2. External Emotion Induction 

The emotion induction was external to the SRT task. Each participant read out six short, emotionally connoting sentences (see Appendix B) and listened to classical music before starting the SRT task [34]. Participants also listened to a classical music extract while performing the task (e.g., Camille Saint-Saëns, Carnival of the Animals for the happiness induction, Igor Stravinsky, Rite of Spring for the neutral induction) [35]. 

AEJE scale (*Echelle d’auto-évaluation de l’état émotionnel du jeune enfant* [Youth Emotional Self-Report Scale]) [36] was used to assess participants’ mood. It is made up of four subscales. Each subscale corresponds to a state of mind (happiness, sadness, fear, or anger) and features five different faces, ranging from the least intense emotion (1 point) to the most intense one (5 points) (see Appendix C). In the present study, participants were each asked to point to the face that best represented their state of mind. Given that the experimental context was nonecological, we only assessed three emotions (happiness, sadness, and fear) [36]. If it was difficult for a child to choose an emotion, the examiner could use pictures to explain the mood intensity represented on the corresponding face. 

### 2.3. Procedure

The present study was conducted between March 2018 and December 2018. Participants were individually tested in a quiet room at their school. The protocol was divided into three parts (see Table 1). First, we assessed the children’s happiness, sadness, and fear on the AEJE scale. The examiner explained, “I am going to show you three scales with some faces on them. The faces on each scale represent different degrees of one state of mind: happiness, sadness, or fear. For each scale, I will ask you to show me the face that best corresponds to your mood right now. There is no right or wrong answer. If you do not know what to choose, I will help you with some pictures”.

Second, each participant read out loud six emotionally connoting sentences and listened to a classical music sample that was either happy or neutral, depending on their subgroup. The participants then evaluated their emotions again on the AEJE scale.

Third, each child performed the SRT task while listening to a different classical music sample through a computer headset. The SRT task lasted 5–10 min. The examiner explained the activity like this: “When you see the star in one of the four rectangles on the screen, press the corresponding key (e.g., “X” for Rectangle A, “C” for Rectangle B, “N” for Rectangle C, and ? for Rectangle D).” There was a short break between each block. When the SRT task was over, the examiner congratulated the participant in order to dispel the induced emotion. All 92 participants completed the SRT task. 

## 3. Results

### 3.1. Emotional Self-Assessment

Participants evaluated their mood on the AEJE scale before (first part) and after (second part) the emotion induction. We report the results as positive emotions (ratings on Happiness scale) and negative emotions (averaged ratings on Fear and Sadness scales). This methodological choice was driven by the fact that before the age of 10, children have difficulty distinguishing between different negative feelings [35]. The mean ratings were analyzed with a repeated-measures general linear model. There are 2 Emotions (Happy and Neutral) × 2 Times factors (before the first part and after the second part of the emotion induction). This allowed us to check the effect of the induction on each participant. Partial eta squared (η^2^p) calculated the effect size of significant main effects and interactions. The effect is small for 0.01 < η^2^p < 0.06; the effect is medium for 0.06 < η^2^p < 0.14; and the effect is large for η^2^p > 0.14.

In the neutral subgroup, there was no significant impact of induction on either the negative, F(1, 26) = 1.3, ns, or positive, F < 1, ns, ratings. In the happy subgroup, there was a significant impact of emotional valence, F(1, 37) = 374, *p* < 0.001, η²p = 0.9. The effect is medium. This effect interacted with time: F(1, 37) = 3.94, *p* = 0.05, η²p = 0.10. The effect is medium. Thus, children’s positive ratings after the happiness induction increased (from 4.29 to 4.37), whereas their negative ratings decreased (from 1.49 to 1.34). In other words, after listening to music and reading positive sentences, children evaluated themselves as being generally happier and less sad or afraid. 

### 3.2. SRT Task

We calculated a repeated-measures general linear model to analyse mean reaction times (in milliseconds) for each block (numbered 1 to 5) and each condition (semi-random or fixed sequence). We decided to eliminate values that fell outside of 2 standard deviations from the mean, that is, when participants responded much too fast or much too slowly compared with the others. Indeed, mean rates can be falsified by these few extreme values. The induction of neutral or happy emotions was included as a between-participants factor. The design of the analysis is as follows: 5 blocks (1–5) × 2 task conditions (fixed, semi-random) × 2 emotions (happy and neutral). 

### 3.3. Reaction Time Analysis 

An overall analysis revealed a significant effect of task condition on reaction times: F(1, 63) = 6.02, *p* < 0.02, η²p = 0.09. The effect is medium. Children were generally faster when the target followed a fixed sequence (974.603 ms) rather than a semi-random one (987.685 ms). However, there was no significant impact of the block, F(4, 252) = 0.962, *p* > 0.05, ns. There was not a significant influence of emotion induction, F(1, 63) = 0.080, *p* > 0.05, ns. However, a descriptive analysis indicated that reaction times differed between the induction of neutral (976.36 ms) and happy (985.928 ms) emotions. Finally, there was a slight three-way interaction between block, condition, and induction, F(4, 252) = 2.328, *p* = 0.05, η²p = 0.04 (the effect is small), indicating different reaction times in relation to condition and emotion induction throughout the task. 

### 3.4. Rates of Correct Responses

An overall analysis highlighted better rates of correct responses during fixed sequences (96.78%) than during random ones (95.8 %), F(1, 63) = 8.169, *p* = 0.006, η²p = 0.12. The effect is medium. The effect of the block was also significant: F(4, 252) = 3.019, *p* < 0.02, η²p = 0.05. The effect is small. Fisher’s post hoc test indicated differences between the first (97.35%), the second (96.22%), the third (95.63%), the fourth (95.86%), and the fifth (96.42%) blocks. The effect of induction was not significant, F > 0.6 ns. 

## 4. Discussion

The purpose of this research is to take a position on the existing controversy about the impact of emotion on implicit learning. Thus, we subjected children aged 8 to 11 years to an SRT task after having induced them emotionally via two inducers. Based on the work of Ellis and Moore [24], we assume that emotion will have a deleterious effect on implicit sequence learning. 

First, we show that children have an implicit learning capacity. The shorter reaction times for fixed sequences than for semi-random ones attest to this type of learning. Higher mean correct response rates in the fixed sequences for all the subgroups also confirmed that implicit learning took place. Indeed, our SRT task is designed such that one semi-aleatory sequence is always directly followed by one fixed sequence. Thus, the participant is not conscious that there are fixed sequences, which increases the reliability of implicit learning [37]. In a highly original way, we find that there was no significant effect of block or significant effect of the interaction between condition and block in any of the subgroups, despite consistently shorter reaction times in the fixed condition. This result goes against our previous work conducted without emotional induction [38]. Thus, the hypothesis that constant emotional re-induction with music throughout the SRT task probably weakened implicit learning is interesting to pose. In particular, it is based on the fact that, although implicit learning is unintentional, it requires careful processing of information [6]. Thus, listening to music with emotional connotations throughout the SRT task certainly mobilized participants’ attentional resources [24] and may have interfered in part with what had been learned implicitly just before.

We show, in a very interesting way, longer reaction times when participants underwent the induction of happiness rather than that of a neutral emotion. This result seems to be in line with the model of attentional resource allocation and cognitive interference according to which emotions could act in the same way as a dual task, consuming cognitive resources [24]. These hypotheses have been extensively tested by Ellis and his colleagues in the field of memory through different recall tasks. Other studies have investigated the effect of the emotional state, positive or negative, on other tasks involving working memory in order to test the hypothesis of resource deprivation and cognitive resources and cognitive interference caused by emotion again (on reasoning tasks [39], on the Tower of London [40], on attentional processes [41]).

## 5. Conclusions

This study allows us to take a position on the specificity of implicit learning. The SRT paradigm allowed us to verify the fact that children were impacted by the emotional nature of the item to be tracked in this type of task. We believe that the induction of constant listening to emotionally charged music throughout the task interfered with learning, resulting in variable reaction times. This interference is dependent on the emotional valence of the music, which is particularly interesting. Indeed, the reaction times of the children are faster the more the music is of positive valence, as if the emotion acts in the same way as a double task, consuming cognitive resources. However, although emotion appears to have an impact on children’s reaction times, it does not appear to influence implicit learning. This result is similar to Sentman’s findings, which show no significant difference in performance on the sequential reaction time task as a function of the subject’s emotional state [32]. In general, our results reinforce Reber’s previously mentioned postulate that implicit learning would be a robust phenomenon [8]. To go further, we can question the nature of the SRT task. Indeed, we have seen that some studies have found differences in performance depending on the paradigm selected (SRT or artificial grammar [32]). The processes involved in these tasks are not the same since the artificial grammar task requires the extraction of probabilistic information, whereas the sequential reaction time task relies on the learning of specific sequences. The model of attentional resource allocation and cognitive interference precisely foresees that the emotional state mobilizes parts of the attentional resources that can no longer be allocated to the task at hand. In this sense, the effect of emotion cannot be observed in the same way in all situations. all situations. The deleterious effect of emotion, i.e., the degradation of performance, should be differentiated according to the attentional cost of the task to be performed. Moreover, these different processes require skills that evolve with age in a non-linear fashion. It is therefore highly likely that developmental differences exist depending on the selected learning task and the emotions induced.

## 6. Future Directions

Regarding our results, it would be interesting to delve deeper into the interaction between emotions and implicit learning in children. Indeed, the impact of emotions on cognitive processes has been widely studied in children, almost exclusively through explicit learning. However, the ability to automatically and implicitly detect certain more or less complex regularities in our environment is a fundamental aspect of human cognition. This implicit learning plays an important role in development. The impact of emotion on implicit processes is not clearly established in adults and has not been explored in children outside our own work. Having answers about the implicit processes more or less impacted by the emotion felt by the child would be a valuable contribution for pedagogues and didacticians and would allow major advances in the field of education.

## Figures and Tables

**Table 1 children-10-00533-t001:** Summary of protocol.

	First Part	Second Part	Third Part
Objectives	1. Mood Assessment	2. Emotion induction 3. Mood Assessment	4. Implicit sequence learning abilities
Tools	1. AEJE scale	2. Emotionally sentences 2. Classical music 3. AEJE scale	4. SRT task

## Data Availability

Data supporting reported results can be found on https://www.nakala.fr/10.34847/nkl.c487cvmr.

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
