# Peer review of "Influence of Emotions on Elementary-School Children’s Implicit Learning during a Serial Reaction Time Task"

_children, 2023, doi:10.3390/children10030533_

Round 1

Reviewer 1 Report

This study investigates the efficiency of implicit learning and the impact of emotion on implicit learning in children aged 8 - 11 years. 

Line 23: Change "...therefore well know both..." to " ...therefore, understand both..."

Line 26: Change "...text understanding." to "...understanding text."

Line 39: Delete "Explicit learning differs from implicit learning."

Line 49: Delete "...up to now..."

Line 50: Delete "...and..."

Line 51: Change "grammars" to grammar"

Line 114: Change "...compared to..." to "...compared with..."

Line 115: This sentence seems to indicate there is a point to be made but nothing follows. Please check it as it seems that there is something missing.

Line 193: Change "...8 11..." to "...8 - 11..." or "...8 to 11..."

Line 220: It may be useful to include the full title of the AEJE and include an English translation.

The first paragraph of the Discussion summarises the study in too much detail. This can be reduced. The second paragraph also restates the findings. I think the first paragraph can be deleted.

Line 301: Change "...aged 8 11 years..." to "...aged 8 to 11 years..."

 The conclusions section doesn't really provide a clear conclusion, rather it speculates about improving the methodology for future research. I suggest writing a clear conclusion that summarises your findings and place the existing paragraph under a "Future Directions" section to follow the Conclusion..

Author Response

Dear reviewer, thank you again for your reading and your expertise, please find attached the answers to your remarks.

Reviewer 2 Report

Line 23 – in a reading or in reading activities

Line 32 – perhaps you can tell us your purpose, aim right here to finish the thought.

Line 46 – I do not think the reference style is using ; between the numbers.

Line 48 – perhaps the subtitle should be Reber’s influence or something along that line.

Line 66 – seems a good place for a new paragraph.

Line 80 – a good place for one more subheading – Criticisms of Implicit Learning Research

Are there not any meta-analyses that fit your topic? I searched a bit. I know the Lochbaum et al. is not your topic. It is in a way your topic concerning emotions and performance and the POMS to performance makes some sense. Again, my question is are there not meta-analyses to cite? Emotions and learning / Emotions and performance are age old topics you know. It seems can find some additional references.

Lochbaum M, Zanatta T, Kirschling D, May E. The Profile of Moods States and Athletic Performance: A Meta-Analysis of Published Studies. European Journal of Investigation in Health, Psychology and Education. 2021; 11(1):50-70. https://doi.org/10.3390/ejihpe11010005

Table 1 does not seem needed. You can add the mean ages in the sentences (line 194).

Line 222 - A figure with the facial responses seems appropriate to present.

A data analysis section is needed in the methods with all planned analyses and explains.

Power analysis?

Line 242 – The SRT task lasted 5 to 10 minutes? Is the to missing?

Maybe Cohen’s d as opposed to partial eta squared? Both are of course effect size values. We need the partial eta squared interpreted.

Table 2 Mood Assessment – assessment does not seem to be capitalized.

Author Response

(The authors gave the same response as above.)

Round 2

Reviewer 1 Report

I thank the Authors for their work and I believe you have sufficiently addressed my concerns regarding your manuscript.

I suggest you delete "Then" in line 329. "We show..."

Author Response

Dear reviewer,
Thank you for reading our article carefully. We have removed the "Then" at your request.
Best regards,
The authors

"We show, in a very interesting way, longer reaction times when participants underwent the induction of happiness, rather than that of a neutral emotion."

Reviewer 2 Report

Thank you for your revision. Some small edits still seem needed, but MDPI does a great job of editing.

line 306 - seems this should be F > 0.60 or something other than 0,6

Table 1 is still difficult to read.

Author Response

Dear reviewer,
Thank you for reading our article carefully. We have changed the wording at your request.
Best regards,
The authors
